# The Molecular Mechanisms of Systemic Sclerosis-Associated Lung Fibrosis

**DOI:** 10.3390/ijms24032963

**Published:** 2023-02-03

**Authors:** Joe E. Mouawad, Carol Feghali-Bostwick

**Affiliations:** 1Division of Rheumatology & Immunology, Department of Medicine, Medical University of South Carolina, Charleston, SC 29425, USA; 2Medical Scientist Training Program, Medical University of South Carolina, Charleston, SC 29425, USA

**Keywords:** systemic sclerosis, scleroderma, fibrosis, fibroblast, lung, pulmonary, interstitial lung disease, molecular mechanisms

## Abstract

Systemic sclerosis (SSc), also known as scleroderma, is an autoimmune disorder that affects the connective tissues and has the highest mortality rate among the rheumatic diseases. One of the hallmarks of SSc is fibrosis, which may develop systemically, affecting the skin and virtually any visceral organ in the body. Fibrosis of the lungs leads to interstitial lung disease (ILD), which is currently the leading cause of death in SSc. The identification of effective treatments to stop or reverse lung fibrosis has been the main challenge in reducing SSc mortality and improving patient outcomes and quality of life. Thus, understanding the molecular mechanisms, altered pathways, and their potential interactions in SSc lung fibrosis is key to developing potential therapies. In this review, we discuss the diverse molecular mechanisms involved in SSc-related lung fibrosis to provide insights into the altered homeostasis state inherent to this fatal disease complication.

## 1. Introduction

Systemic sclerosis (SSc), or scleroderma, is an autoimmune connective tissue disease with one of the highest mortality rates among the rheumatic diseases [1]. Fibrosis is recognized to be a defining feature of SSc, affecting the skin and multiple visceral organs [2]. As a result, SSc is considered the prototypic fibrosing disease. For more than 20 years, interstitial lung disease (ILD), characterized by lung fibrosis, has been the leading cause of death in SSc [3,4]. This is largely due to the lack of treatments that can stop or reverse the fibrotic process. Currently, only two drugs are approved by the Food and Drug Administration (FDA) for SSc, but these merely reduce the progression of ILD rather than stop or reverse it [5,6]. SSc thus remains incurable due to progressive lung fibrosis, with lung transplantation as the only viable option, which is impossible on the scale that it is needed [7]. 

The molecular mechanisms mediating SSc-related lung fibrosis are complex and incompletely understood. While the final outcome is the excessive deposition of extracellular matrix (ECM) resulting in lung fibrosis, it is evident that the pathways leading to this outcome are numerous, involving different molecular and cellular components [8]. Understanding the different pathogenic mechanisms that contribute to lung fibrosis in SSc and their interplay is key to identifying potential molecular targets for therapy. In this review, we describe the different molecular mechanisms currently implicated in SSc-related lung fibrosis in an effort to establish a well-needed comprehensive source for better understanding the disease pathogenesis.

## 2. Gene Expression Profile of SSc

High-throughput gene expression studies on SSc lung tissues and cells have proven to be valuable in identifying molecular pathways underpinning the pathogenesis of SSc-associated lung fibrosis. A microarray analysis performed on normal versus explanted SSc lung tissues and matching primary pulmonary fibroblasts revealed that the latter showed differentially expressed genes corresponding to ECM components and fibrotic signaling molecules as well as novel genes and pathways that were not previously reported in the SSc lung. Molecular signatures included those that were unique to SSc and those that correlated with the fibrotic phenotype in SSc and idiopathic pulmonary fibrosis (IPF) [9,10]. Pro-fibrotic and ECM signatures were also identified in an independent microarray analysis of SSc lung biopsies [11]. This latter study also identified a macrophage activation signature and interferon signatures [11]. It is worth noting that Hsu et al.’s analysis focused on SSc lungs with primary usual interstitial pneumonia on pathology [9], while Christmann et al. included lung tissues with non-specific interstitial pneumonia on pathology [11]. Moreover, Tyler et al. identified a three-gene network of interacting alleles in a cohort of 416 SSc patients, namely, *WNT5A, RBMS3,* and *MS12,* which influenced lung outcomes in SSc. Gene expression profiling has also focused on the fibroblasts in the lung, as they are the effector cells in fibrosis [12,13]. A bulk RNA sequencing (RNA-seq) analysis of fibroblasts derived from healthy versus SSc lung tissues revealed an expression profile similar to that of whole lung tissues, with increased expression of ECM genes, including *COL1A1* and *COL3A1*, and fibrotic genes such as *TGFB2, IGF2, IGFBP3, IGFBP5,* and *WNT5A*, while antifibrotic genes such as *MMP1, MMP9, CTSL, SFRP1*, and *IL33* were underexpressed in SSc lung fibroblasts [14]. Taken together, these findings suggest that fibroblasts are the predominant source of the aberrant lung tissue expression profile identified using microarray analysis. These findings also emphasize the notion that fibrosis is a dynamic process caused by the disruption of a balance to favor pro-fibrotic factors over anti-fibrotic pathways. More recently, single-cell RNA-seq (scRNA-seq) technology has proven to be a powerful tool to garner insights into individual cellular contributions to lung fibrosis in SSc [15]. Valenzi et al. demonstrated, using scRNA-seq, that SSc lung tissues show a unique heterogeneity in fibroblast populations, namely, SPINT2^hi^, MFAP^hi^, WIF^hi^, and a new myofibroblast population with high *ACTA2* expression, all showing differential expression profiles when compared with the control [13]. The data reinforced that myofibroblast differentiation and proliferation are key drivers of disease pathogenesis [13]. Similarly, using scRNA-seq, Reyfman et al. showed differential gene expression in the fibroblast subpopulations of fibrotic lungs that included but were not limited to SSc-ILD [16]. Their analysis also identified a novel population of pro-fibrotic alveolar macrophages, distinct epithelial cell signatures, and novel airway stem cells that were across fibrosing lung diseases [16]. In activated fibroblasts, the differential expression of ECM and pro-fibrotic genes including *COL1A1*, *COL3A1*, *POSTN, TAGLN*, and *ACTA2* was also noted, validating findings of microarray and RNAseq studies [17]. These findings further reinforce the established disease paradigm that fibroblasts are the driving force behind lung tissue fibrosis in SSc. Thus, understanding the various molecular mechanisms of fibroblast activation and proliferation in SSc is essential for mapping the pathology of the disease.

## 3. Deregulated Molecular Pathways in SSc

### 3.1. Fibrotic Factors

#### 3.1.1. Transforming Growth Factor Beta (TGFβ)

TGFβ is one of the most widely studied pro-fibrotic factors in the context of fibrosis. The TGFβ superfamily includes TGFβ, bone morphogenetic proteins (BMPs), growth/differentiation factors (GDFs), activins, and inhibins [18]. TGFβ plays a crucial role in transitioning fibroblasts into activated myofibroblasts, which are responsible for the excessive production of ECM in fibrosis [19]. TGFβ signals via interaction with two receptor serine/threonine kinases, known as type I and type II receptors, which form a heterotetrameric complex upon ligand binding [20]. Upon complexing, the autophosphorylation of type I and II receptors mediates the docking and phosphorylation of Smad 2/3, which in turn interact with Smad 4 to create a transcriptional complex that translocates to the nucleus and activates or represses multiple target genes [21]. While TGFβ works mainly via activation of the Smad pathway, it can also activate other non-canonical pathways [18]. In fibroblasts, TGFβ signaling shifts the gene expression profile to a profibrotic phenotype, inducing the expression of profibrotic and ECM genes, while suppressing the antifibrotic and matrix-degrading genes, leading to tissue fibrosis [19,22]. TGFβ can also exert similar effects on other cell types, such as epithelial and endothelial cells, and can induce their transition into alpha smooth muscle actin-expressing myofibroblasts [19,23].

In SSc, TGFβ-regulated genes are differentially expressed in the fibrotic lungs of patients, which is correlated with the severity of the disease [24]. This is consistent with the notion that TGFβ plays a central role in SSc pathology [25]. In fact, we previously demonstrated that fibroblasts derived from the lungs of SSc patients express higher TGFβ1 and TGFβ2 levels than fibroblasts from healthy lungs [26]. This supports previous findings obtained by Christmann et al. indicating that lung tissues from SSc patients show amplified expression of genes regulated by TGFβ [11]. Recent evidence has shown that macrophages polarized to the alternatively activated phenotype (M2) are also a major source of TGFβ [27]. A follow-up study showed that the expression of TGFβ by M2 macrophages is amplified by methyl-CpG-binding domain 2 (MBD2) protein, which suppresses the expression of an inhibitor upstream of TGFβ, and MBD2 was found to be overexpressed in SSc-ILD lung tissues [28]. Zehender et al. recently demonstrated a novel mechanism for TGFβ-induced fibrosis in SSc, which involves a loss of epigenetic control over autophagy via a Smad3-dependent downregulation of the H4K16 histone acetyltransferase MYST1, mediating the activation of fibroblasts [29]. Core regulators of autophagy, BECLIN1 and ATG7, were consequently found to be upregulated in SSc dermal fibroblasts, as well as fibrotic skin and lungs of mice overexpressing TGFBRI, while their knockdown alleviated fibrosis [29].

Efforts to target TGFβ as a therapeutic strategy to reduce SSc lung fibrosis have not been effective, and concerns about potential adverse complications due to the pleiotropic roles of TGFβ in lung physiology have led to efforts focusing on targeting other pro-fibrotic factors and molecular pathways as a therapeutic strategy to treat SSc [1,30,31]. 

It is worth noting that several members of the TGFβ family have been largely overlooked in SSc lung research to date, although they are likely to play important roles in disease pathogenesis. Unlike TGFβ, whose active form is generated when and where it is needed, activin A and BMP4 are generally readily active and thus possess distinct signaling dynamics from TGFβ-induced fibrosis [32].

#### 3.1.2. Platelet-Derived Growth Factor (PDGF)

PDGF has also been shown to play a central role in organ fibrosis, since stromal mesenchymal cells, including fibroblasts, express PDGF receptor isoforms that are activated and drive processes implicated in fibrosis, such as proliferation, migration, and ECM deposition [33]. In fact, lung fibrosis of various etiologies, whether environmental exposure, transplant rejection, autoimmune, or idiopathic, have been associated with increased PDGF levels in bronchoalveolar lavage fluid (BALF) or lung tissues [34]. There are two PDGF receptor isoforms, PDGFRα and PDGFRβ, which are tyrosine kinase receptors recognized by four ligand isoforms, PDGF-A, PDGF-B, PDGF-C, and PDGF-D [35]. Upon ligand binding, the homo- or hetero-dimerization of the receptors leads to autophosphorylation events of their cytoplasmic domain, which activates downstream signaling pathways, including phosphatidylinositol 3 kinase (PI3K), Ras-MAPK, Src, and phospholipase Cγ (PLCγ) pathways [35]. More recently, we showed that PDGF can also signal via melanin-concentrating hormone receptor 1 (MCHR1), modulating intracellular cyclic adenosine monophosphate (cAMP) production and inducing TGFβ1 and connective tissue growth factor (CTGF) expression in fibroblasts, thus promoting a fibrotic response [36]. 

PDGF-A and PDGF-B levels are elevated in the BALF of SSc patients [37]. Interestingly, SSc-derived fibroblasts exhibit unique, positive cross-talk between PDGF and TGFβ signaling, which does not occur in healthy fibroblasts [38]. In addition, microRNA miR-30b, which suppresses the expression of PDGFRβ, is downregulated in the serum of SSc patients [39]. Reducing the expression of PDGFRβ in SSc dermal fibroblasts with miR-30b inhibited collagen synthesis and myofibroblast activation [39]. Since miR-30b levels are decreased in the circulation of SSc patients, it is reasonable to extrapolate these findings to lung fibroblasts. Recently, Svegliati et al. demonstrated that PDGF and anti-PDGFR autoantibodies, which are elevated in SSc patient serum [40,41], stimulated higher growth rate, migration, and expression of collagen in human pulmonary artery smooth muscle cells, which was attributed to the generation of reactive oxygen species, and elevated NOX4 and mammalian target of rapamycin (mTOR) [42]. All these findings have made PDGF an attractive molecular target for therapeutic treatment of SSc lung fibrosis [43]. In fact, nintedanib, a drug that blocks the ATP-binding pocket of PDGFR and other tyrosine kinase receptors, such as fibroblast growth factor receptor (FGFR) and vascular endothelial growth factor receptor (VEGFR), was approved by the FDA for the treatment of SSc-ILD [6,44]. At the cellular level, nintedanib blocks the PDGF-induced differentiation of lung fibroblasts into myofibroblasts, reduces their proliferation and migration, and suppresses the expression of collagen and fibronectin, supporting the antifibrotic outcome of blocking PDGF signaling.

#### 3.1.3. Fibroblast Growth Factor (FGF)

FGFs are a family of signaling proteins that can act in an endocrine, paracrine, or even intracrine manner [45]. Under paracrine or endocrine conditions, target cells interact with FGF ligands via four receptor tyrosine kinases, FGFR1, FGFR2, FGFR3, and FGFR4 [45]. Intracrine FGFs are nonsignaling, in that they act independently of FGFRs and mainly serve as cofactors for voltage-gated sodium channels [45,46]. Upon ligand binding, FGFRs activate multiple pathways, including PI3K, Ras-MAPK, PLCγ, and STAT signaling pathways [45]. The roles of the different FGF ligands in fibrosis have been variable in the experimental models, with some promoting lung fibrosis and others protecting against it [47]. For example, members of the FGF family of proteins can activate fibroblasts and induce their proliferation and ECM deposition [48]. In contrast, one member of the family, FGF19, was found to be protective against lung fibrosis in mice, and its levels were decreased in the plasma of IPF patients [49]. However, studies about the specific role of FGFs in SSc-related lung fibrosis are scarce. Recently, Chakraborty et al. demonstrated a mechanistic involvement of FGF9 and its receptor FGFR3 in SSc, both of which are upregulated in SSc fibrotic skin [50]. FGF9 was shown to bind FGFR3 and activate dermal fibroblasts from SSc skin, leading to the downstream stimulation of AKT, p38, extracellular signal-regulated kinase (ERK), and calcium/calmodulin-dependent protein kinase 2 (CAMK2) and promoting cyclic adenosine 3′,5′-monophosphate response element binding protein (CREB) activation, which induced the expression of profibrotic mediators [50]. These findings from SSc skin have not yet been validated in tissues or primary cells derived from lungs of SSc patients.

#### 3.1.4. Wnt/β-Catenin Signaling

Widely known for its role in organ and tissue development, the Wnt/β-catenin signaling pathway has more recently been implicated in fibrotic disorders in different organs [51,52,53,54]. The binding of Wnt ligands to their Frizzled (Fz) receptors triggers downstream effects inhibiting the degradation of β-catenin, stabilizing it in the cytoplasm, and promoting its translocation to the nucleus, where it results in the transcription of Wnt target genes [55]. Earlier studies have confirmed the involvement of the Wnt/β-catenin pathway in the pathogenesis of lung fibrosis and, specifically, in SSc [56,57,58]. In addition, SSc skin fibroblasts express high levels of Wnt proteins such as Wnt1 and Wnt10b, coupled with decreased expression of the Wnt antagonists SFRP1, DKK1, and WIF1 [59]. Our group recently reported similar results in SSc lung fibroblasts, showing increased Wnt5a and decreased SFRP1 expression in SSc lung fibroblasts compared with normal lung fibroblasts [14]. More recently, increased levels of a novel isoform of CD146 that activates myofibroblasts were noted in the serum of SSc patients with pulmonary fibrosis, a process driven by Wnt5a [60]. This is consistent with previous studies showing that the Wnt/β-catenin pathway is activated in SSc lung fibrosis, allowing its downstream pro-fibrotic effects to be promoted [57,61]. 

#### 3.1.5. Interleukins

Interleukins (ILs) are a group of cytokines with immunoregulatory functions known to be secreted by white blood cells, but also several other cell types, such as epithelial and stromal cells [62,63]. There are more than 40 distinct ILs, each eliciting different functions across multiple different cell types via binding to high-affinity receptors [64]. Several of these ILs have been shown to directly interact with fibroblasts to promote lung fibrosis [65]. 

IL-6 has been extensively studied in the context of SSc, and its levels are increased in SSc serum, skin, and fibroblasts [66,67,68,69]. Serum levels of IL-6 strongly correlate with the severity of SSc lung fibrosis and are predictive of mortality in SSc patients, suggesting a profibrotic role of IL-6 in lung tissues [70]. Our group confirmed a significant increase in IL-6 expression in lung tissues derived from SSc patients compared with healthy lung tissues [10] and showed that the increase in lung fibroblasts is mechanistically driven, at least in part, by upregulated lysyl oxidase (LOX) via activation of c-Fos [71]. Tocilizumab, a monoclonal antibody targeting the IL-6 receptor, preserved lung function in SSc patients, slowing lung fibrosis progression, and was approved by the FDA for the treatment of SSc-ILD [5,72,73].

The involvement of multiple other ILs in SSc-related lung fibrosis has been documented. Yaseen et al. demonstrated a significant increase in IL-31 in SSc serum coupled with the overexpression of its receptor, IL-31RA, in SSc-derived lung fibroblasts, and confirmed the profibrotic effects of IL-31 in mouse lungs [74]. Moreover, IL-11 was significantly upregulated in SSc lung fibroblasts [75,76]. Follow-up studies confirmed that IL-11 drives the activation of lung fibroblasts. Specifically, mice with conditional fibroblast-specific knockout of the IL-11 receptor were protected from bleomycin-induced lung fibrosis, and IL-11 knockout lung fibroblasts were refractory to TGFβ stimulation [77]. IL-1 induces miR-155, which was overexpressed in SSc lung fibroblasts, leading to increased TGFβ and collagen synthesis, as well as further feed-forward expression of IL-1 driven by inflammasome activation [78]. Other interleukins whose serum levels correlated with ILD severity in SSc patients include IL-13, IL-17, IL-33, and IL-34; however, their exact roles in promoting lung fibrosis are incompletely delineated [79,80,81,82].

#### 3.1.6. Insulin-Like Growth Factors (IGFs) and Their Binding Proteins (IGFBPs)

IGFs and IGFBPs have been implicated in the pathogenesis of SSc lung fibrosis. IGF-I levels are increased in the BALF of SSc patients, and the protein itself promotes the proliferation of fibroblasts [83]. IGF-II expression is also increased in fibrotic SSc lung tissues and fibroblasts and induces ECM deposition via the activation of PI3K and Jun N-terminal kinase pathways [84]. A follow-up study demonstrated that IGF-II signaled via type 1 IGF receptor (IGF1R), insulin receptor (IR), and a hybrid IGF1R/IR complex receptor, promoting fibrosis through multiple mechanisms: by directly activating myofibroblasts, by increasing ECM production while reducing its degradation, and by stimulating the expression of TGFβ2 and TGFβ3 [26]. We showed that IGFBP-3 and IGFBP-5 are profibrotic proteins that stimulate the production of ECM by lung fibroblasts [85,86,87]. IGFBP-5 is significantly overexpressed in SSc lung tissues and fibroblasts and induces the expression of collagen type I, fibronectin, CTGF, LOX, and DOK5 [88,89]. Mice expressing human IGFBP-5 showed sustained increased expression of ECM genes [90]. Recently, IGFBP-2 serum levels were found to have prognostic value for assessing the development of ILD in SSc patients, but further studies are needed to understand its potential role in lung disease [91].

#### 3.1.7. YAP/TAZ

Yes-associated protein (YAP) and transcription coactivator with PDZ-binding motif (TAZ) are key components of the Hippo pathway [92]. Mammalian STE20-like (MST) and large tumor suppressor kinase (LATS) are the core proteins upstream of YAP/TAZ in the Hippo pathway [93]. When the Hippo pathway is activated, the sequential phosphorylation of MST, LATS, and then YAP/TAZ occurs, causing the retention of YAP/TAZ in the cytoplasm [94]. YAP/TAZ signaling mediates its effect via cross-talk with multiple other pathways, including the TGFβ and Wnt/β-catenin pathways [94]. YAP/TAZ was shown to promote myofibroblast proliferation, contraction, and ECM synthesis [95]. Myofibroblast-specific YAP/TAZ deficiency ameliorates fibrosis across multiple organs, including lungs, kidneys, and liver, supporting previous findings on the critical role that YAP/TAZ signaling plays in fibrogenesis [96]. Further delineation of the role of YAP/TAZ signaling in SSc-related fibrosis is warranted. Toyoma et al. demonstrated that targeting YAP/TAZ with dimethyl fumarate is a viable therapeutic strategy for dermal fibrosis in SSc [97]. More recently, Wu et al. showed that skin fibroblasts and serum from SSc patients had increased levels of YAP and TAZ compared with healthy controls, and they demonstrated that knockdown of YAP/TAZ in mice alleviated bleomycin-induced lung fibrosis [98]. 

#### 3.1.8. Chemokines

Chemokines are a family of small proteins that play a critical role in maintaining homeostasis and regulating inflammation and tissue-specific leukocyte migration [99]. Upon tissue damage, chemokines initiate and maintain the inflammatory process, and the fine-tuning of their expression is imperative for the resolution of inflammation, culminating in tissue repair and wound healing [100]. Chemokines can activate fibroblasts and perpetuate their ECM production and deposition [100]. This has led to research on the roles of certain chemokines in SSc lung disease. Specifically, C-C motif ligand-2 (CCL2) levels were elevated in the BALF of SSc patients compared with healthy controls [101]. Furthermore, CCL2 protein was overexpressed in fibroblasts from fibrotic lungs of SSc patients [102]. Wu et al. demonstrated that CCL2 plasma levels can serve as a biomarker and a potential therapeutic target for ILD progression in SSc patients, as higher CCL2 levels predicted a faster decline in forced vital capacity (FVC%) over time [103]. Another chemokine, CXCL4, was reported by van Bon et al. to be elevated in SSc patients and correlated with the severity of lung fibrosis [104]. A follow-up study by Affandi et al. demonstrated that CXCL4 is required for bleomycin-induced lung fibrosis in mice, where it promoted myofibroblast transformation, leading to excessive ECM deposition [105]. Multiple other chemokines have been implicated in SSc lung fibrosis, such as CCL5, CCL7, CCL18, CXCL3, and CXCL8, mostly as potential biomarkers of ILD [100,106,107]. Further studies are still needed to elucidate which of these chemokines are central to the pathogenesis of SSc lung fibrosis.

### 3.2. Anti-Fibrotic Factors

While most studies focus on the pro-fibrotic molecular contributors to SSc lung fibrosis, reduced levels and/or blunted activity of anti-fibrotic pathways constitute a crucial component of uncontrolled fibrosis. Here, we describe anti-fibrotic proteins and pathways whose reduced expression or activity has been implicated in SSc lung fibrosis. 

Cathepsins are lysosomal proteases widely known for their role in intracellular housekeeping for the maintenance of cellular homeostasis, such as antigen processing and the degradation of proteases and chemokines [108]. However, cathepsins are not limited to the lysosome and have been shown to play important extracellular functions, particularly in remodeling and degrading the ECM [109,110]. Cathepsin S (CTSS) levels were significantly decreased in the serum of SSc patients compared with healthy controls, and the reduced levels were reflective of the severity of SSc lung fibrosis [111]. More recently, we reported that Cathepsin L (CTSL) expression was significantly reduced in lung fibroblasts and tissues derived from SSc patients, in part due to constitutive suppression by the TGFβ/Smad pathway, and that the lack of availability of CTSL in the extracellular milieu prevented the release of the anti-fibrotic protein endostatin [14]. Endostatin is released by proteolytic cleavage from the C-terminus of collagen-XVIII and has shown potent anti-fibrotic effects on human fibroblasts and human tissues as well murine models of fibrosis [112]. Endostatin is in fact detected in the circulation and BALF of SSc patients and patients with other lung fibrosing diseases, but its levels do not reach therapeutic levels, suggesting a blunted anti-fibrotic response in SSc-associated fibrosis [113,114].

Matrix metalloproteinases (MMPs) are ECM-degrading enzymes that play an important role in maintaining tissue matrix homeostasis [115]. We showed, using bulk RNAseq, that the expression of multiple MMPs, including MMP1, MMP9, MMP15, and MMP28, is reduced in lung fibroblasts derived from SSc patients compared with healthy controls [14]. MMP19-deficient mice had an augmented lung fibrotic response to bleomycin when compared with wildtype mice, and their lung fibroblasts overexpressed collagen type I and alpha smooth muscle actin, a marker of myofibroblast activation [116,117]. In addition, E4, an anti-fibrotic peptide derived from the C-terminus of endostatin, activates the urokinase pathway in primary lung fibroblasts and lung tissues, leading to the induction of MMP1 and MMP3 and consequently promoting matrix degradation and fibrosis resolution [118].

Multiple other proteins have been shown to play a protective role against SSc-related lung fibrosis. For example, Chu et al. demonstrated that deacetylase Sirtuin 1 was underexpressed in the peripheral blood mononuclear cells (PBMCs) of SSc patients with lung fibrosis, and its activation in bleomycin-treated mice or overexpression in human lung fibroblasts reduced collagen production and ameliorated the fibrotic response [119]. Sirtuin 3 has also been implicated in SSc skin and lung fibrosis [120]. Further, decreased serum levels of Sirtuins 1 and 3 correlated with lung fibrosis in SSc [121]. The expression of other sirutins, notably, Sirtuin 7, is also decreased in SSc lung fibroblasts, resulting in increased Smad3 levels [122]. As we note above, two members of the IGFBP family, IGFBP-3 and -5, are increased in SSc skin and lung and promote fibrosis [85,86,87]. In contrast, another member of the family, IGFBP-4, is underexpressed in SSc lung fibroblasts and exhibits anti-fibrotic activity via the suppression of CTGF and C-X-C chemokine receptor 4 (CXCR4) [123]. Antagonism of Wnt signaling is another anti-fibrotic pathway. The expression of secreted frizzled proteins (SFRPs) 1 and 4, which are Wnt antagonists, was suppressed upon bleomycin administration in mice via the hypermethylation of the promoter region, which induced Wnt overactivity and progression of lung fibrosis [124]. Another Wnt antagonist, DKK1, was also suppressed via the hypermethylation of its gene promoter in SSc fibroblasts [125]. This is supported by findings showing that methyl cap binding protein-2 (MeCP2), which methylates the SFRP1 and DKK1 promoter regions, is overexpressed in SSc fibroblasts, resulting in reduced expression of its target genes [126,127]. In addition, miR27a-3p, which targets SFRP1 expression, is upregulated in SSc fibroblasts, leading to reduced SFRP1 levels in the circulation of SSc patients [128]. While the latter studies were mainly focused on SSc dermal fibroblasts, our group confirmed the reduced expression of SFRP1 and DKK1, coupled with Wnt overexpression, in SSc lung fibroblasts, supporting the protective role of these Wnt antagonists also in SSc lung fibrosis [14]. Taken together, the reduced expression of anti-fibrotic genes in SSc-associated lung fibrosis supports the concept of a blunted or suppressed anti-fibrotic response in SSc.

### 3.3. Extracellular Molecules as Pro- and Anti-Fibrotic Mediators

In addition to increased production of ECM components and the enzymes responsible for their crosslinking in fibrosis, matricryptins have emerged as important players in fibrosis. Matricryptins are cleavage products of extracellular matrix proteins and glycosaminoglycans that exert biological activity. For example, endostatin is cleaved from Collagen XVIII and exerts anti-fibrotic effects [112]. In contrast, endotrophin, released from Collagen VI, has pro-fibrotic effects [129]. In addition, enzymes extensively studied for their matrix crosslinking function can harbor additional functions relevant to fibrosis. As an example, LOX has moonlighting functions that include the induction of IL-6 and the increased production of ECM components [71]. Thus, approaches to mitigate lung fibrosis in SSc should take into consideration the levels and activity of the various enzymes that cleave molecules in the ECM, resulting in the release of matricryptins [130], and those implicated in ECM stabilization that have additional novel functions in promoting fibrosis.

## 4. Conclusions

ILD remains the leading cause of death in SSc due to the lack of effective treatments that halt or reverse fibrosis. Understanding the multitude of molecular pathways involved in SSc-driven lung fibrosis and building a comprehensive view of their interconnections are critical to identify effective targets for anti-fibrotic therapies. The dynamic nature of fibrosis is the result of an abundance of pro-fibrotic factors and paucity of anti-fibrotic ones (Figure 1). While most therapeutic approaches to SSc-ILD have focused on targeting profibrotic molecular pathways, strategies to induce the endogenous anti-fibrotic response could serve as a viable and more effective therapeutic strategy, since endogenous molecules are less likely to demonstrate toxicity or elicit an autoimmune response. The end goals are to restore homeostasis, halt the progression lung fibrosis, and even reverse it, thus reducing mortality in SSc and other fibrosing lung diseases such as IPF.

## Figures and Tables

**Figure 1 ijms-24-02963-f001:**
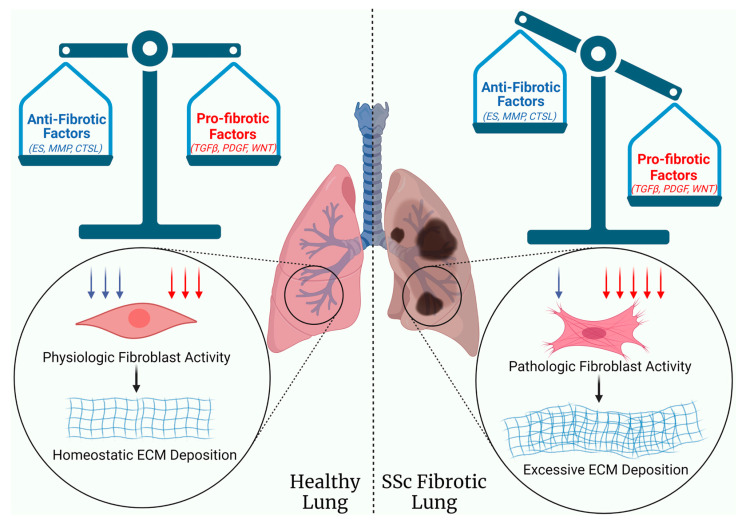
Lung fibrosis in SSc is the result of an imbalance between anti-fibrotic and pro-fibrotic factors, favoring the latter, and resulting in the activation of fibroblasts and the excessive deposition of ECM. ES, endostatin; MMP, matrix metalloproteinase; CTSL, Cathepsin L; TGFβ, Transforming Growth Factor Beta; PDGF, Platelet-Derived Growth Factor; WNT, Wingless-Related Integration Site; ECM, extracellular matrix; SSc, systemic sclerosis.

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
