# Peer review of "The Molecular Mechanisms of Systemic Sclerosis-Associated Lung Fibrosis"

_ijms, 2023, doi:10.3390/ijms24032963_

Round 1

Reviewer 1 Report

The article „The Molecular Mechanisms of Systemic Sclerosis-Associated Lung Fibrosis is well-organized and written paper. However, the article needs some corrections.

- Abstract: The sentence „...leads in mortality rate among the rheumatic diseases” needs correction.

- Lines 63-64: The authors wrote „and fibrotic signaling genes such as TGFB2, IGF2, IGFBP3, IGFBP5, and WNT5A, 62 while antifibrotic genes such as MMP1, MMP9, CTSL, and TIMP1 were under-expressed 63 in SSc lung fibroblasts ”, and cited the article: Mouawad, J.E.; Sharma, S.; Renaud, L.; Pilewski, J.M.; Nadig, S.N.; Feghali-Bostwick, C. Reduced Cathepsin L Expression and 408 Secretion into the Extracellular Milieu Contribute to Lung Fibrosis in Systemic Sclerosis. Rheumatology (Oxford) 2022, 409 doi:10.1093/rheumatology/keac411.

However, the referred article did not include information about MMP and TIMP. Please check if this is a correct citation. Moreover, why TIMPs (Tissue inhibitors of metalloproteinase) are called anti-fibrotic (e.g. PMID: 17026855)?

- Please refer also to Tyler et al., PMID: 31694854

Author Response

We would like to thank the reviewers for their careful and thoughtful review. We have modified the draft to include the reviewers’ suggestions and corrections.

Reviewer 1:

The article „The Molecular Mechanisms of Systemic Sclerosis-Associated Lung Fibrosis ” is well-organized and written paper. However, the article needs some corrections.

- Abstract: The sentence „...leads in mortality rate among the rheumatic diseases” needs correction.

Response: The sentence is now corrected. 

- Lines 63-64: The authors wrote „and fibrotic signaling genes such as TGFB2, IGF2, IGFBP3, IGFBP5, and WNT5A, 62 while antifibrotic genes such as MMP1, MMP9, CTSL, and TIMP1 were under-expressed 63 in SSc lung fibroblasts ”, and cited the article: Mouawad, J.E.; Sharma, S.; Renaud, L.; Pilewski, J.M.; Nadig, S.N.; Feghali-Bostwick, C. Reduced Cathepsin L Expression and 408 Secretion into the Extracellular Milieu Contribute to Lung Fibrosis in Systemic Sclerosis. Rheumatology (Oxford) 2022, 409 doi:10.1093/rheumatology/keac411.

However, the referred article did not include information about MMP and TIMP. Please check if this is a correct citation. Moreover, why TIMPs (Tissue inhibitors of metalloproteinase) are called anti-fibrotic (e.g. PMID: 17026855)?

Response: We thank the reviewer for the careful review. The information and data are included in figure 5A of the referenced paper, although we agree with the reviewer that it is not the main focus of the manuscript. We apologize for the error regarding TIMP1 and thank the reviewer for catching it. TIMP1 was removed and replaced with other established antifibrotic genes found downregulated in the figure of the referenced paper.

- Please refer also to Tyler et al., PMID: 31694854.

Response: We thank the reviewer for suggesting the reference. It is now added in lines 58-60.

Reviewer 2 Report

In this review, Mouawad and Feghali-Bostwick have summarized molecular mechanisms in lung fibrosis induced by systemic sclerosis (SSc) along with related etiologies. Overall, this review is well-written and captures key molecular players in fibrosis. Discussion of literature is up to date. I have a few suggestions/comments as outlined below.

1.     Is TIMP1 considered an antifibrotic gene? (Line 63). I think most studies report TIMP1 as a fibrotic gene as it is upregulated in tissues from various fibrotic models.

2.     Discussion regarding markers used to established distinct fibroblast populations from scRNA-seq studies would add molecular insights for potential roles of each cell population in fibrosis (Line 68-80).

3.     Elaboration of evidence that other members of the TGFbeta superfamily could be involved in SSc lung fibrosis could strengthen this argument (Line 123-125).

4.     The authors may consider adding detailed discussion regarding the conflicting roles of FGFs in lung fibrosis as they could shed light on SSc-induced fibrosis (Line 169-170).

5.     Typos for “were” not “where” in line 221 and 222?

6.     How does miR-155 induce expression of IL-1? (Line 223) Is it a direct target? If so, miR-155 should suppress expression of IL-1.

7.     Adding examples of key pro- and anti-fibrotic factors into the figure could be helpful for readers as a summary. Alternatively, a table of summary of pro- and anti-fibrotic factors could also work.

Author Response

We would like to thank the reviewers for their careful and thoughtful review. We have modified the draft to include the reviewers’ suggestions and corrections.

Reviewer 2:

In this review, Mouawad and Feghali-Bostwick have summarized molecular mechanisms in lung fibrosis induced by systemic sclerosis (SSc) along with related etiologies. Overall, this review is well-written and captures key molecular players in fibrosis. Discussion of literature is up to date. I have a few suggestions/comments as outlined below.

  1. Is TIMP1 considered an antifibrotic gene? (Line 63). I think most studies report TIMP1 as a fibrotic gene as it is upregulated in tissues from various fibrotic models.

Response: We appreciate the reviewer’s careful review. The reviewer is correct. We apologize for the oversight. TIMP1 is now removed from the text and replaced with other established antifibrotic genes.

  1. Discussion regarding markers used to established distinct fibroblast populations from scRNA-seq studies would add molecular insights for potential roles of each cell population in fibrosis (Line 68-80).

Response: We thank the reviewer for the suggestion. Clarification of the fibroblast population markers used in the scRNA-seq studies was added to the text.

  1. Elaboration of evidence that other members of the TGFbeta superfamily could be involved in SSc lung fibrosis could strengthen this argument (Line 123-125).

Response: We thank the reviewer for the suggestion. Information relating to other members of the TGFβ family and their potential involvement in fibrosis is added in the text.

  1. The authors may consider adding detailed discussion regarding the conflicting roles of FGFs in lung fibrosis as they could shed light on SSc-induced fibrosis (Line 169-170).

Response: We thank the reviewer for the suggestion. The information is now added.

  1. Typos for “were” not “where” in line 221 and 222?

Response: We thank the reviewer for catching the typos. We have made the corrections.

  1. How does miR-155 induce expression of IL-1? (Line 223) Is it a direct target? If so, miR-155 should suppress expression of IL-1.

Response: We apologize for the mistake. IL-1 induces miR-155, not the opposite as we had stated. This is now corrected and further clarified in the text.

  1. Adding examples of key pro- and anti-fibrotic factors into the figure could be helpful for readers as a summary. Alternatively, a table of summary of pro- and anti-fibrotic factors could also work.

Response: We thank the reviewer for the suggestion. We added examples of key pro-and anti-fibrotic factors to the figure.

Round 2

Reviewer 1 Report

The authors have adequately addressed the reviewers' comments.

However, there is a minor issue that the authors should explain the abbreviations used in the Figure in the Figure legend, as the Figure should be self-explanatory.

Author Response

We would like to thank the reviewer for the suggestion. We have clarified in the figure legend the abbreviations used in the figure.